# Effect of Die Configuration on the Physico-Chemical Properties, Anti-Nutritional Compounds, and Sensory Features of Legume-Based Extruded Snacks

**DOI:** 10.3390/foods10123015

**Published:** 2021-12-05

**Authors:** Michela Costantini, Martins Sabovics, Ruta Galoburda, Tatjana Kince, Evita Straumite, Carmine Summo, Antonella Pasqualone

**Affiliations:** 1Department of Soil, Plant and Food Science (DISSPA), University of Bari Aldo Moro, Via Amendola, 165/A, I-70126 Bari, Italy; carmine.summo@uniba.it; 2Department of Food Technology, Latvia University of Life Sciences and Technologies, Rigas Iela 22, LV-3004 Jelgava, Latvia; martins.sabovics@llu.lv (M.S.); ruta.galoburda@llu.lv (R.G.); tatjana.kince@llu.lv (T.K.); evita.straumite@llu.lv (E.S.)

**Keywords:** extrusion-cooking, legumes, extruder die, texture profile, phytates, oligosaccharides

## Abstract

Legumes are not valued by all consumers, mostly due to the prolonged soaking and cooking process they require. This problem could be solved by preparing legume-based ready-to-eat snacks. In this study, the effect of two different dies (circular and star-shaped, with cross-sections of 19.6 and 35.9 mm^2^, respectively) on the physico-chemical properties, anti-nutritional compounds, and sensory features of extruded breakfast snacks was determined. Extruded products were obtained from 100% legume flours of red lentil, faba bean, brown pea, and common bean. The extrusion-cooking conditions were 2.5 g/s feed rate; 160 ± 1 °C die temperature; 16 ± 1 g/100 g feed moisture, and 230 rpm screw speed. Star-shaped extrudates showed a lower expansion ratio, degree of starch gelatinization, and water solubility index, as well as higher bulk density, hardness, crunchiness, and lightness (*L**) values. The oligosaccharides showed non univocal variations by changing the die, whereas phytates did not vary at all. The extrudates from lentil flour (richer in fiber) were the least preferred by sensory panelists, due to their hard texture. However, the spherical extrudates were preferred over the star-shaped product. These results emphasize the possibility of improving the physico-chemical and sensory properties of legume extrudates by selecting a proper die.

## 1. Introduction

Appreciated by vegetarians, vegans, and suitable for coeliac patients, legumes play an important role in human health due to their nutritional composition [1]. Rich in proteins and complex carbohydrates, legumes are also characterized by their low-fat content [2]. Furthermore, they are an excellent source of dietary fiber, B-group vitamins and minerals [3]. However, legumes preparation is a time-consuming step, mainly due to the prolonged soaking and cooking process these grains require. This problem, which reduces their consumption by consumers, could be solved by developing new food products, such as legume-based ready-to-eat snacks [1].

Extrusion-cooking is a low cost, multifunctional and versatile food processing technique, which subjects raw materials to heat, pressure, and shear forces causing, among major biochemical effects, starch gelatinization, protein denaturation, fiber degradation, amylose-lipid complex formation, and Maillard reaction [4]. This technique is largely used for the development of ready-to-eat expanded snacks with different shapes, textures, and colors, characterized by enhanced flavor [5].

In recent years extensive research has been carried out to develop extruded products from legumes [6,7], in most cases blended with cereals [1,8,9,10]. Specific studies evaluated the effect of extrusion-cooking parameters, such as feed moisture, extrusion temperature, and screw speed on the nutritional [11], physico-chemical [8,10], textural [12,13], and sensory characteristics [14] of the final product, also by means of trials directly carried out at industrial level [15]. However, no studies were carried out on the effect of the die, which determined the shape and size of the finished products during the extrusion of the legumes.

Food shape and size are particularly important in capturing consumer attention [16]. These characteristics strongly influence the implicit associations that individuals make regarding an object and its value [17]. In addition, shape and size may significantly influence the physico-chemical characteristics [18] and even the sensory properties of food [19].

The aim of this study was, therefore, to determine the effect of two different die configurations (circular and star-shaped, with cross-sections of 19.6 and 35.9 mm^2^, respectively) on the physico-chemical properties, anti-nutritional compounds, and sensory features of extruded breakfast snacks prepared from 100% legume flour (red lentil, faba bean, brown pea, and common bean).

## 2. Materials and Methods

### 2.1. Materials

Dehulled legume flours from four different species, specifically red lentil (*Lens culinaris* Medik.), faba bean (*Vicia faba* L.), brown pea (*Pisum sativum* L.), and common bean (*Phaseolus vulgaris* L.) were used to obtain extruded snacks. Red lentil flour was purchased from a local market, the faba bean and brown pea flours were supplied by Ltd. Aloja-Starkelsen (Ungurpils, Latvia), and the common bean flour was supplied by the Priekuli Research Centre (Institute of Agricultural Resources and Economics, Priekuli district, Latvia). All flours were sieved on a 0.25 mm screen. The nutritional characteristics of these flours (as reported on the labels) are shown in Table 1.

### 2.2. Flour Conditioning

Flours were conditioned to reach the optimal moisture content for extrusion (16 g/100 g), as determined in the preliminary experimental tests. Considering the moisture content of each flour type (12.95 ± 0.01, 10.87 ± 0.01, 12.54 ± 0.06 and 11.46 ± 0.06 g/100 g for red lentil, faba bean, brown pea, and common bean flour, respectively), the amount of water to be added was calculated. Water was then progressively added to flour in a dough mixer (BEAR Varimixer AR10, Wodschow & Co., Brondby, Denmark) at a medium speed, to avoid the formation of lumps, until an evenly hydrated flour was obtained (about 20 min).

### 2.3. Extrusion-Cooking Process

The extrusion-cooking process was carried out using a DSE30 Lab Twin-screw extruder (Jinan Sunward Machinery Co., Ltd., Jinan City, China) with an extrusion capacity of 12 kg/h. The extruder, which had a 5-kW motor, was equipped with two 38CrMoAl screws (32 mm diameter, 660 mm length, 500 rpm maximum screw speed). The operating conditions, selected in the preliminary experimental tests, were as follows: feed rate, 2.5 g/s; barrel temperature of the three heating zones = 55, 95, and 125 °C, respectively; die temperature, 160 °C; feed moisture, 16 g/100 g; screw speed, 230 rpm. Two different dies, with a circular and a star-shaped hole, were used (Figure 1). The cross-section of the circular die nozzle was 19.6 mm^2^, whereas the star cross-section area accounted for 35.9 mm^2^. Both die holes had a length of 6.35 mm. To analyze the water absorption index, water solubility index, starch gelatinization degree, color, phytate and oligosaccharides content, the extrudates were ground by using an electric grinder HM-5735 (Hoomei Electrical Appliance Co., Monza, Italy), to pass through a 0.25 mm sieve. The other analyses were carried out on the entire extrudates.

### 2.4. Bulk Density and Expansion Ratio

Bulk density (BD) of the extruded products was determined by a rapeseed displacement method, and calculated according to the Equation (1) reported by Koksel and Masatcioglu [20]. Randomly selected whole pieces (10 ± 0.1 g) of each type of extruded product were weighted:BD (g/cm^3^) = We × Ve^−1^ = We × (ρs × Ws^−1^)(1)
where, We and Ve are the weight (g) and the equivalent volume (cm^3^) of the extruded products, respectively. Ve coincides with the ratio between the rapeseed density (ρs) and the rapeseed weight (Ws), with the same volume as the extrudates. Five replicates for each sample were carried out.

The expansion ratio (ER) was calculated as the ratio of extruded product diameter (measured using a calliper) to extruder die hole diameter, as reported by Koksel and Masatcioglu [20]. Ten replications were carried out.

### 2.5. Water Absorption Index and Water Solubility Index

The water absorption index (WAI) and the water solubility index (WSI) of the extruded products were determined according to Equations (2) and (3), reported by Janve and Singhal [21].
WAI (g/g) = (weight of sediment)/(sample weight)(2)
WSI (g/100 g) = [(weight of dry solids in supernatant)/(sample weight)] × 100(3)

The WSI is the weight of dry solids in the supernatant, whereas WAI is the weight of sediment without the supernatant per unit weight of the sample analyzed. The determination was carried out in triplicate.

### 2.6. Degree of Starch Gelatinization (DG)

The degree of starch gelatinization (DG) of the extruded products was determined using the method reported by Liu et al. [22], based on the formation of a blue iodine complex with amylose released during gelatinization, with slight modifications. Then, 40 mg of sample was dissolved in 50 mL of 0.15 M KOH, the suspension was mixed for 15 min, and then centrifuged for 10 min at 4032× *g* to remove the insoluble sediment. After centrifugation, 1 mL of supernatant was neutralized with 9 mL of 0.017 M HCl. Subsequently, 0.1 mL of iodine reagent (prepared by dissolving 1 g iodine and 4 g potassium iodine in 100 mL of water) was added to the neutralized solution. After mixing, the absorbance was measured at 600 nm (A1) using a Cary 60 UV–VIS spectrophotometer (Agilent Technologies Inc., Santa Clara, CA, USA). For each sample, a control was prepared by using 1 M KOH and 0.1 M HCl instead of KOH (0.15 M) and HCl (0.017 M). The average of three measurements was taken and the DG was computed using the Equation (4):DG = A1/A2(4)
where A1 and A2 are the absorbance at 600 nm of sample and control, respectively.

### 2.7. Texture Analysis

Texture analysis was carried out using the TA.HD. Plus texture analyzer (Stable Microsystems Ltd., Godalming, UK) equipped with a cylindrical probe, with a diameter of 4.5 cm and a back extrusion cell with an inner diameter and height of 5.0 and 7.0 cm (Stable Microsystems Ltd.), respectively. The cell was filled with 100 mL of extruded products and the sample was compressed to 50% of the original height, as described by Smith and Hardacre [23]. The test was conducted at the following conditions: 1.0 mm/s pre-test speed; 5.0 mm/s test speed; 10 mm/s post-test speed; 0.049 N trigger force; 1 kN load cell. Hardness value was considered as the maximum compression force and expressed in Newtons (N). Crispiness was the number of positive peaks in the force versus time graph, while crunchiness (N·s) was the linear distance of the rugged lines obtained from the same graph. A greater number of positive peaks indicated a greater number of fracture events, thus, crisper extrudates. A longer linear distance resulted in a longer drop from the peak for each fracture event on average, and thus, resulted in crisper extrudates [24]. Data acquisition was performed using the “Exponent” software (Stable Microsystems Ltd.). Five replications for each sample were carried out.

### 2.8. Bowl Life Analysis

The bowl life analysis was carried out using the method reported by Oliveira et al. [25], with slight modifications. Briefly, 100 mL of extruded products were soaked in milk (fat content 2.5 g/100 g) at 5 °C for 3 min, and then drained for 10 s. Subsequently, texture analysis of the milk-soaked extruded products was performed under the same conditions as described above for the dry products. Five replications were carried out.

### 2.9. Color Determination

The color of the flours and the extruded products were determined using the CM-600d colorimeter (Konica Minolta Sensing Inc., Osaka, Japan) equipped with the SpectraMagic NX software (Konica Minolta, Tokyo, Japan). Lightness (*L**), redness (*a**), and yellowness (*b**) were determined. Five replications were carried out.

### 2.10. Determination of Total Phytates Content

Total phytate content of flours and extruded products was determined according to the method reported by Summo et al. [26]. In order to express the phytate content in the sample, expressed as mg/g of phytic acid on dry matter, the results were multiplied by 0.282 (molar ratio of phytate-phosphorus in a molecule of phytate). Three replications were carried out.

### 2.11. Determination of Oligosaccharides

Oligosaccharides (verbascose, stachyose and raffinose) of flours and extruded products were determined by high-performance liquid chromatography (HPLC) (Agilent Technologies, Santa Clara, CA, USA), equipped with a 300 × 7.8 mm cation exchange column (Rezex RCM column, Ca^2+^, 8 μm, Torrance, CA, USA) and Refractive Index Detector (RID 1260, Agilent Technologies), as previously reported in De Angelis et al. [27] with few modifications. Then, 10 mg of flour or ground sample were dispersed in 10 mL of deionized water, stirred for 5 min and filtered through 0.22 µm cellulose acetate filter. The HPLC separation was conducted isocratically at a flow rate of 0.8 mL/min, a column temperature of 80 °C and a RID temperature of 40 °C. Deionized water was used as the mobile phase. The identification was carried out by comparing the retention time with that of the corresponding standard (Merck KGaA, Darmstadt, Germany). A calibration curve for each oligosaccharide was prepared for the quantification. The analysis was carried out in triplicate and the results were expressed as mg/g of each oligosaccharide on dry matter.

### 2.12. Sensory Evaluation

Twenty-eight semi-trained panelists from the Faculty of Food Technology, Latvia University of Life Sciences and Technologies (Jelgava, Latvia) evaluated the liking of legume-based extruded snacks according to a ranking test [28]. The samples, coded with random numbers and arranged in pieces of 3 on transparent glass plates, were randomly placed on the tray to be served to each panelist at the same time. For taste neutralization between samples, warm black tea was used. Panelists were asked to arrange the extruded product samples from 1 (the most-liked sample) to 8 (the least-liked sample), according to their degree of liking for four sensory attributes (appearance, texture, taste, and aftertaste), using the evaluation form generated by Fizz Acquisition 2.51 software (Biosystems, Couternon, France). The obtained data were reported as sum of ranks for each sample.

### 2.13. Statistical Analysis

The experimental data of the legume flours and the extruded products were subjected to one-way ANOVA and two-way ANOVA, respectively, followed by the Tukey’s HSD test. The two-way ANOVA analysis was carried out considering the type of legume flour and the type of die as factors. Significant differences among the values of all parameters were determined at *p* < 0.05 by the Minitab 17 Statistical Software (Minitab, Inc., State College, PA, USA, 2010). Data obtained from the sensory evaluation were statistically analyzed by the Friedman test using Fizz Calculation 2.60 software (Biosystems, Couternon, France), resulting in a significance level set at *p* < 0.05.

## 3. Results and Discussion

### 3.1. Characteristics of Flours Used in the Experiments

The flours used to produce the extruded snacks are shown in Figure 2. Significant differences in the color parameters (*L**, *a** and *b**) were found among them (Table 2). Red lentil flour had the highest *a** and *b** values and the lowest *L** value. Common bean flour was the lightest, followed by brown pea and faba bean flours. Brown pea flour showed the lowest *a** and *b** values, the latter without a significant difference with common bean flour.

Significant differences (*p* < 0.05) among flours were observed in the content of anti-nutritional compounds (Table 3). Legumes contain several anti-nutritional compounds, including phytic acid and non-digestible oligosaccharides [29,30]. Phytates chelate several important divalent cations, such as Fe, Zn, Ca, and Mg, reducing their availability for absorption and use in the small intestine [31]. Raffinose family oligosaccharides, such as raffinose, verbascose and stachyose, cause flatulence and discomfort in humans [32].

Common bean flour had the highest content of phytates, followed by faba bean, brown pea, and red lentil flours, respectively, in agreement with other studies [33]. Brown pea flour was characterized by the highest verbascose, although it had the lowest stachyose content. Faba bean flour had the highest content of stachyose and the lowest content of raffinose. The content of oligosaccharides may vary among different legume species and varieties and depends on the growing environment [34]. Vidal-Valverde et al. [35] reported a high variability among 18 different varieties of pea for raffinose (4.10–10.30 mg/g), stachyose (10.70–26.7 mg/g) and verbascose (0.00–26.70 mg/g). Tahir et al. [36] found higher values of stachyose than raffinose and verbascose in 11 lentil varieties, in agreement with our findings. Oligosaccharides represent a major limitation for the extensive use of legumes, both at a domestic and an industrial level [37].

### 3.2. Physico-Chemical Properties of Extruded Products

Significant differences (*p* < 0.05) were found among the extruded products for all physico-chemical parameters, except for water absorption index (WAI), as a function of the *legume* used, the type of *die*, and their interaction *(legume × die*) (Table 4). WAI was not influenced by the type of die.

Considering the legume type, the extruded products obtained from red lentil had the highest bulk density (BD) and WAI, as well as the lowest expansion ratio (ER) and water solubility index (WSI). Brown pea, instead, showed the highest ER. The spherical extrudates of brown pea, in particular, were well expanded and larger than others with the same shape (Figure 3). Common bean extrudates, both spherical and star-shaped, showed the highest degree of starch gelatinization (DG). All flours tended to expand more through the circular die, leading to extruded products characterized by higher ER, DG, and WSI, as well as a lower BD, than the star-shaped ones.

The effect of the die could be explained by considering the different friction to which the product was subjected during the extrusion. The circular cross-section was smaller than the star-shaped one (19.6 vs. 35.9 mm^2^). Consequently, the conditioned flour was subjected to elevated levels of friction and pressure flowing through the circular die, compared with the star-shaped one. In turn, higher levels of friction induced heat generation and increased the actual extrusion temperature. Higher pressure and temperature are known to promote more expanded and less dense products [11,38], explaining the higher ER and lower BD of spherical extrudates. In addition, a specific effect of geometry can be hypothesized, not related to the size of the die hole. The star-shaped cross section is characterized by the presence of angles, absent in the circular one, which could have caused a mechanical breaking of bubbles in the gelatinized starchy matrix flowing through the die, further disturbing the expansion of the extrudate. The increase in the die nozzle diameter was found to cause a decrease in radial expansion in yellow corn extrudates [39]. A higher extrusion pressure also induces a higher degree of starch gelatinization [39], as was observed in the spherical extrudates, compared with the star-shaped ones, except for the red lentil-based products, due to their high fiber content. The presence of the fiber, in fact, restricted the starch gelatinization required for the expansion of the expanded snacks [40]. Starch gelatinization, in turn, positively influenced the volume expansion of the extrudate.

ER and BD are physical parameters capable of influencing the consumer acceptability of extruded products [11]. Several studies reported an inverse relation between BD and ER [8,11]. The same trend was observed in our work, where a negative correlation between BD and ER (*r* = −0.631; *p* = 0.093) was found. Furthermore, the fiber present in the starting flour may also affect both of these parameters, as reported by other researchers [41,42,43]. Dietary fibers lead to cell-wall rupture, before air bubbles can expand, reducing the overall expansion [42]. As a result, extruded products with a high fiber content are usually compact and hard, not crispy, and have an undesirable texture [44]. Therefore, red lentil flour, showing the highest fiber content (Table 1), led to extrudates with the lowest ER and highest BD.

The extrusion-cooking conditions related to the die used, as the presence of fibers could also influence the WAI and WSI values, representing the amount of water absorbable by the extruded product, and the quantity of soluble substances formed during the extrusion process from starch, proteins, and fibers [21].

WSI was influenced by all factors considered (legume, die and legume × die); however, the effect of die on WAI was found not significant. Regarding the effect of the type of legume on WAI, higher fiber levels, absorbing and retaining the water within a well-developed starch-protein-polysaccharide network, resulted in an increase in this parameter, as reported by Tas and Shah [45]. Furthermore, extrusion-cooking may induce structural modifications, such as the reduction in the fiber particle size increasing the surface area and, therefore, their water absorption capacity [46]. Although extrusion-cooking is known to induce an increase in the WSI parameter, due to the degradation of polymers to low molecular weight soluble compounds [47], other factors, such as the interaction between fiber and starch, might have affected WSI [48]. In fact, red lentil products, which had the highest total dietary fiber content, were characterized by the lowest WSI. WSI, which was influenced by the die configuration, was higher in spherical than in star-shaped extrudates, due to the increase in extrusion temperature induced by friction, in the case of the circular die.

### 3.3. Texture of Dry and Milk-Soaked Extruded Products

The texture analysis (Table 5) was not applicable to the red lentil extrudates, due to their particularly hard texture (confirmed by their high BD and low ER), exceeding the instrumental range of measure of the texture analyzer used. Other authors observed that extrudates with high BD and low ER were characterized by very large [21] to non-analyzable hardness [49]. A significant effect (*p* < 0.05) of the type of *legume*, *die* configuration and their interaction (*legume* × *die*) was observed for all textural parameters, except for crispness of dry products, which was not influenced by the die. Star-shaped extrudates were characterized by higher hardness and crunchiness than the spherical extrudates, in agreement with the ER and BD values. As for crispness, brown pea spherical extrudates were the crispiest, without significant differences to the brown pea star-shaped extrudates. This trend was due to the higher ER and lower BD and hardness values, which characterized both brown pea products. Koksel and Masatcioglu [20] reported a significant negative correlation between ER and hardness, as well as between BD and crispness, in yellow pea puffs. Other researchers reported that extrudates which were less hard and crunchy had higher crispness values [25,50]. Crispness and crunchiness are important quality attributes, used to describe the texture of extruded snack products. Consumer acceptability is strongly influenced by both crispness [51] and crunchiness [50,52]. Crispness and crunchiness are two sensations that in the human brain are induced by different stimuli, during the dynamic process of mastication [53]. Crispness could be identified as the perceived force necessary to separate the product into two or more distinct pieces during a single bite with the incisors [53]. Crisp products are characterized by a brittle and low-density structure, which easily breaks and generates loud and high-pitched sounds when fractured [54]. Crunchiness is the perceived intensity of repeated incremental failures of the product during a single complete bite with molar [53] teeth. Crunchy foods exhibit harder textures and emit sounds at lower frequencies than crisp foods [54].

The obtained results may be explained again considering the effect of die size. A larger die cross section may reduce the viscosity and the capacity of mechanical energy dissipation inside the extruder, producing harder and less crispy products [25]. To assess the textural quality in the real conditions of consumption, the bowl life test was performed, by soaking the extrudates in milk. Usually, expanded products have a greater number of pores, which reduces the resistance to mass transfer and increases the rate of water absorption [55]. Milk, however, contains some fat, which may block the pores of the extrudates, reducing the absorption rate [55]. All the textural parameters decreased after soaking, due to milk absorption and consequent softening. Liquid uptake, indeed, modifies the microstructure and the mechanical strength of the extruded products by plasticizing and softening the starch and protein matrix [51]. Soaking caused a hardness reduction of 20, 35 and 50% for brown pea, faba bean, and common bean spherical extrudates, respectively. The star-shaped products behaved similarly, with a reduction of hardness ranging from 47 to 62% for brown pea and common bean extrudates, respectively. Crunchiness and crispness decreased even further, by 76% and 74% on average. The values of crispness agreed with those of other studies [56].

### 3.4. Color of Extruded Products

Color is one of the most important characteristics of a food product, due to its marked influence on consumer acceptability [20]. Color features are known to be influenced by the extrusion-cooking [7]. All extrudates were darker than the corresponding flours (Figure 2). Their overall appearance is shown in Figure 4. All color components were significantly influenced by *legume* type, *die* configuration and *legume* × *die* interaction (Table 6). Color features of the examined products were the result of existing pigments and the partial incidence of Maillard reaction, due to the extrusion-cooking conditions.

All star-shaped products were characterized by higher *L** and lower *a** values (the latter with the exception of red lentil) than spherical ones. The *b** index showed a non-univocal trend. The temperature rises and shear stress related to the rotation of the screw during the extrusion can degrade pigments, especially carotenoids, with consequent color alterations [57]. The increase of *a** and the decrease of *L** may be also associated with the formation of brown polymers, namely melanoidins, during the Maillard reaction [12]. An increase of *b** could be due to the formation, during the initial stages of the Maillard reaction, of yellow-colored compounds, or due to the thermal oxidation of lipids in the sample [12]. A die with a larger cross section results in lower pressure, and lower heat development during extrusion [38,58], leading to a less intense Maillard reaction. The consequences are a less pronounced browning and a reduced flavor development [11]. Therefore, the star-shaped extrudates, obtained through a larger cross-section which made the extrusion process less drastic, were lighter than the spherical ones.

### 3.5. Anti-Nutritional Compounds of Flour and Extruded Products

The extrusion-cooking processing conditions, and the type of raw material adopted, can both affect the amount of anti-nutritional compounds found in legume extrudates [7,59]. Comparing the native flours (Table 3) with the extrudates (Table 7), a different behavior was found for different anti-nutritional compounds. For both star-shaped and spherical products, phytates decreased during the extrusion-cooking of the faba bean (12% on average), common bean (23.5% on average), and brown pea (7.9% on average) flours, probably due to the thermal treatment related to the extrusion-cooking process.

Ciudad-Mulero et al. [60] reported a greater reduction of total phytates in lentil flour extruded at 160 than at 140 °C. Oligosaccharides, on the contrary, increased, especially stachyose, and raffinose. This result could be due to the high temperature and pressure adopted during the extrusion-cooking process, which can break the bonds between oligosaccharides and other macromolecules, or may change the structure of the food matrix, improving the extractability of these compounds [59]. The same findings were reported by other researchers in extruded lentil snacks and in pea-rice gluten free expanded products [59,61].

Significant differences (*p* < 0.05) were found for verbascose, stachyose and raffinose as a function of the type of legume, die configuration, and their interaction (*legume* × *die*), whereas phytic acid was not influenced by the die. Red lentil spherical extrudates were characterized by significantly higher verbascose content than star-shaped ones, whereas raffinose increased by approximately 7% in the latter. Common bean stars had higher stachyose and raffinose content than spheres obtained from the same flour. In particular, considering the spherical products, a reduction of both stachyose and raffinose by 1.7 and 7.5% was found, respectively. No significant differences were found between the two die shapes in the faba bean and brown pea extrudates, for all the oligosaccharides. Therefore, the behavior of single oligosaccharides depended on the extrusion conditions employed which, in turn, were related to the size and shape of the die cross-section, but also greatly depended on the type of legume considered. A higher extrusion temperature and pressure may induce the hydrolysis of verbascose into raffinose and stachyose, increasing their contents [61], as found in the brown pea extrudates—both spherical and star-shaped—as well as in the common bean star-shaped extrudates. Specifically, in both brown pea spherical and star-shaped products, a reduction in verbascose of approximately 4% and 3.2%, was observed, respectively. Instead, an increase in both stachyose (by 15 and 17%) and raffinose (by 13 and 9.7%) for spherical and star-shaped products, were found, respectively. Although common bean star-shaped extrudates showed the same behavior as brown pea products, raffinose increased more (16%) than stachyose (5.5%). However, Borejszo and Khan [62] reported a decrease in raffinose and stachyose content in pinto bean flour extruded at 163 °C, whereas the content of all sugars analyzed in an extruded product containing pea increased with the extrusion process [59]. Overall, and with more evidence for raffinose, we observed a higher reduction of oligosaccharides in the spherical products, obtained with a die which induced higher pressure and heat generation than the star-shaped one.

### 3.6. Sensory Evaluation of the Extruded Products

The ranking test results showed that both the flour and die used had a significant influence (*p* < 0.05) on the liking of the extruded product sensory attributes (appearance, texture, taste, and aftertaste) (Table 8). Extrudates obtained from red lentil (particularly the star-shaped) were the least liked for all sensory attributes, with the worst rank sum for “texture”. This result was due to their hard structure (too hard to be analyzed instrumentally with the texture analyzer), being difficult to chew, and their bland taste and aftertaste. However, red lentil spherical extrudates were liked for “appearance” and “aftertaste”, similarly to spherical and star-shaped extrudates from faba bean and brown pea, as well as the star-shaped common bean. Altaf et al. [5], studying chickpea-rice-extruded snacks, reported that higher values of BD and hardness can make the product undesirable for the consumer. In another study, 100% lentil extrudates were less accepted than extrudates from blends of red lentil and corn [63]. On the other hand, common bean extrudates, particularly the spherical ones, were most liked for all the attributes considered, being properly puffed with a crunchy structure, pleasant taste and aftertaste.

Overall, the spherical extrudates were more appreciated than the star-shaped ones. Shape was an intrinsic factor, able to influence the consumer perception and acceptability of the food products. It could even influence taste perception [19,64]. However, spherical and star-shaped extrudates did not show a significant difference in taste and aftertaste, probably because the difference in textural features, i.e., appearance and structure, prevailed.

## 4. Conclusions

The results obtained represent a step forward in the attempt to understand the effect of die configuration during the extrusion of legumes. Most of the dies used in the industry have a circular cross section, which are therefore are commonly used in research investigations, whereas a star-shaped die had not been studied on legumes.

The configuration of the die significantly influenced the physico-chemical properties and sensory features of the legume-based extruded breakfast snacks. In particular, the use of the star-shaped die, with a larger cross-section, resulted in products with a lower ER and higher BD than the spherical extrudates, probably because of lower friction during extrusion. Furthermore, a lower extrusion pressure also induced a lower degree of starch gelatinization in the star-shaped extrudates, compared with the spherical ones, except for the red lentil-based products. The effect of the die on WAI was found insignificant, whereas WSI was higher in the spherical, rather than in the star-shaped extrudates, due to the increase of heat generation induced by elevated friction and pressure flowing through the circular die.

Spherical extrudates were characterized by higher crispiness, and lower hardness and crunchiness than the star-shaped extrudates. The brown pea spherical products were the crispiest. Moreover, the spherical products were characterized by higher *L** and lower *a** values (the latter with the exception of red lentil) and were more appreciated by panelists than the star-shaped extrudates. Regarding the anti-nutritional compounds, the oligosaccharides showed non univocal variations by changing the die, whereas phytates did not vary at all.

Considering that the type of legume also showed a significant influence on the qualitative and nutritional features of the extrudates—presumably related to the fiber content of the flour—the increased knowledge on the effect of the die configuration could be useful for maximizing the expansion of legume-based raw materials, in order to meet consumer expectations for healthy food products with pleasant sensory properties.

## Figures and Tables

**Figure 1 foods-10-03015-f001:**
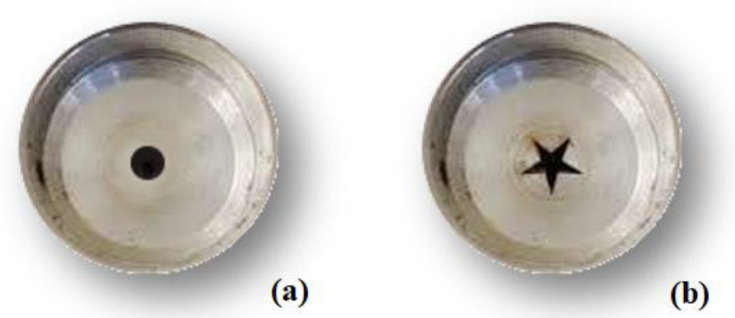
Circular (**a**) and star-shaped (**b**) dies used in the experimental trials.

**Figure 2 foods-10-03015-f002:**
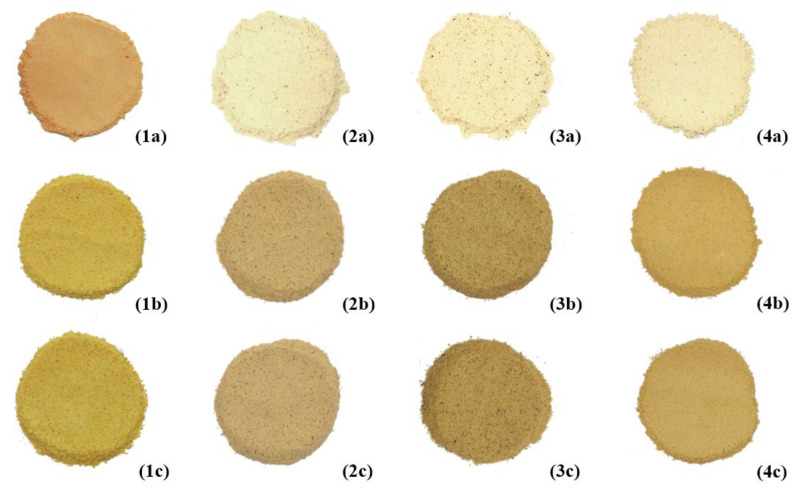
Color of the legume-based extrudates. From left to right: flours and ground extrudates of red lentil (**1**), faba bean (**2**), brown pea (**3**), and common bean (**4**). The letter “a” indicates the flours and the letters “b” and “c” indicate ground spherical and star-shaped extruded products, respectively.

**Figure 3 foods-10-03015-f003:**
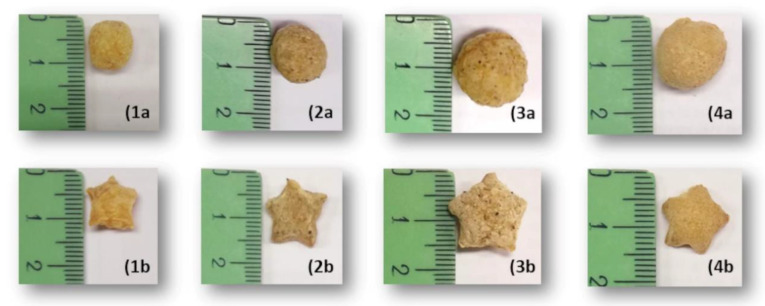
Size of the legume-based extrudates. From left to right: extrudates of red lentil (**1**), faba bean (**2**), brown pea (**3**) and common bean (**4**). The letters “a” and “b” indicate spherical and star-shaped extruded products, respectively.

**Figure 4 foods-10-03015-f004:**
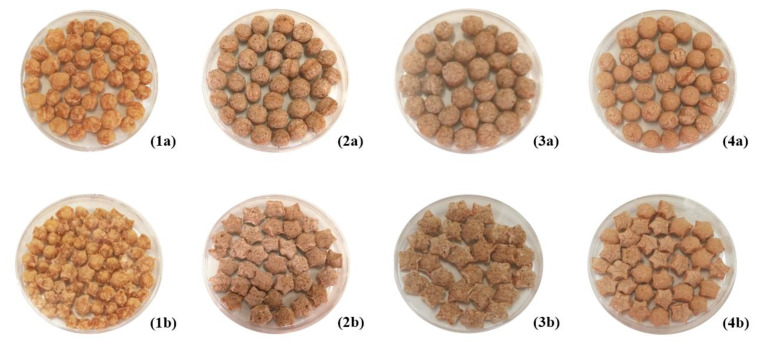
Overall appearance of the legume-based extrudates. From left to right: extrudates of red lentil (**1**), faba bean (**2**), brown pea (**3**) and common bean (**4**). The letters “a” and “b” indicate the spherical and star-shaped extruded products, respectively.

**Table 1 foods-10-03015-t001:** Proximate composition and energy value of red lentil, faba bean, brown pea and common bean flours. All values are expressed on fresh matter.

	Red Lentil	Faba Bean	Brown Pea	Common Bean
Fats (g/100 g)	1.10	1.39	1.97	1.51
Carbohydrates (g/100 g)	29.50	58.90	47.80	49.88
Total dietary fibers (g/100 g)	30.50	8.40	10.50	15.50
Proteins (g/100 g)	25.80	17.80	25.10	21.40
Energy value (kcal/100 g)	292.10	336.00	330.00	329.71

**Table 2 foods-10-03015-t002:** Color parameters (mean ± standard deviation, *n* = 5) of flours of red lentil, faba bean, brown pea, and common bean.

	Red Lentil	Faba Bean	Brown Pea	Common Bean
*L**	84.79 ± 0.36 ^c^	87.56 ± 0.03 ^b^	87.85 ± 0.31 ^b^	89.12 ± 0.37 ^a^
*a**	9.45 ± 0.12 ^a^	−0.49 ± 0.03 ^b^	−0.75 ± 0.03 ^c^	−0.56 ± 0.02 ^b^
*b**	21.85 ± 0.25 ^a^	14.50 ± 0.05 ^b^	13.16 ± 0.02 ^c^	12.93 ± 0.04 ^c^

Different letters in the rows indicate significant differences (*p* < 0.05) among legume flour.

**Table 3 foods-10-03015-t003:** Anti-nutritional compounds (mean ± standard deviation, *n* = 3) of flours of red lentil, faba bean, brown pea, and common bean.

	Red Lentil	Faba Bean	Brown Pea	Common Bean
Phytates (mg phytic acid/g d.m.)	3.49 ± 0.09 ^d^	7.51 ± 0.09 ^b^	4.69 ± 0.02 ^c^	9.09 ± 0.11 ^a^
Verbascose (mg/g d.m.)	16.09 ± 0.09 ^b^	12.74 ± 0.10 ^c^	29.66 ± 0.23 ^a^	12.65 ± 0.40 ^c^
Stachyose (mg/g d.m.)	28.89 ± 0.67 ^c^	39.32 ± 1.11 ^a^	19.16 ± 0.85 ^d^	35.37 ± 0.98 ^b^
Raffinose (mg/g d.m.)	16.28 ± 0.68 ^a^	2.80 ± 0.14 ^d^	10.10 ± 0.88 ^b^	5.99 ± 0.43 ^c^

Different letters in the rows indicate significant differences (*p* < 0.05) among legume flour.

**Table 4 foods-10-03015-t004:** Physico-chemical parameters (mean ± standard deviation) of spherical and star-shaped extruded products obtained from different legume flours (*n* = 5 for BD; *n* = 10 for ER; *n* = 3 for the other parameters).

Die Configuration	Legume	Physico-Chemical Parameter				
		BD (g/cm^3^)	ER	WAI (g/g)	WSI (g/100 g)	DG (g/100 g)
Spherical	Red lentil	0.41 ± 0.01 ^b^	2.10 ± 0.22 ^c^	3.83 ± 0.18 ^a^	9.32 ± 0.30 ^d^	95.21 ± 0.63 ^c^
	Faba bean	0.21 ± 0.02 ^d^	2.46 ± 0.09 ^b^	3.22 ± 0.05 ^b^	16.79 ± 0.25 ^a^	98.29 ± 0.20 ^a^
	Brown pea	0.21 ± 0.00 ^d^	2.78 ± 0.12 ^a^	3.34 ± 0.14 ^b^	12.99 ± 0.89 ^b^	96.96 ± 0.66 ^b^
	Common bean	0.20 ± 0.01 ^d^	2.56 ± 0.23 ^b^	2.49 ± 0.09 ^c^	12.95 ± 0.61 ^b^	98.05 ± 0.05 ^a^
Star-shaped	Red lentil	0.60 ± 0.03 ^a^	1.14 ± 0.04 ^f^	4.05 ± 0.20 ^a^	7.68 ± 0.36 ^e^	96.82 ± 0.21 ^b^
	Faba bean	0.27 ± 0.02 ^c^	1.56 ± 0.05 ^de^	2.60 ± 0.03 ^c^	13.08 ± 0.56 ^b^	93.25 ± 0.14 ^d^
	Brown pea	0.27 ± 0.01 ^c^	1.70 ± 0.17 ^d^	3.33 ± 0.02 ^b^	10.62 ± 0.20 ^cd^	92.75 ± 0.40 ^d^
	Common bean	0.25 ± 0.01 ^c^	1.42 ± 0.07 ^e^	2.81 ± 0.18 ^c^	11.77 ± 0.61 ^bc^	96.99 ± 0.01 ^b^
Legume		*p* < 0.001	*p* < 0.001	*p* < 0.001	*p* < 0.001	*p* < 0.001
Die		*p* < 0.001	*p* < 0.001	*p* = 0.659	*p* < 0.001	*p* < 0.001
Legume × die		*p* < 0.001	*p* < 0.05	*p* < 0.001	*p* < 0.01	*p* < 0.001

BD = bulk density; ER = expansion ratio; WAI = water absorption index; WSI = water solubility index; DG = degree of starch gelatinization. Different letters in column indicate significant differences (*p* < 0.05) between both shapes of products (sphere and star) considering the interaction between the two factors (*legume* and *die*).

**Table 5 foods-10-03015-t005:** Texture (mean ± standard deviation; *n* = 5) of dry and milk-soaked spherical and star-shaped extruded products obtained from different legume flours.

Die Configuration	Legume	Dry			Milk-Soaked (Bowl Life)		
		Hardness (N)	Crunchiness (N.s N·s)	Crispness	Hardness (N)	Crunchiness (N·s)	Crispness
Spherical	Red lentil	n.d.	n.d.	n.d.	n.d.	n.d.	n.d.
	Faba bean	606 ± 7 ^c^	1799 ± 84 ^c^	51.3 ± 2.1 ^cd^	395 ± 3 ^a^	610 ± 10 ^a^	14.3 ± 2.9 ^b^
	Brown pea	477 ± 13 ^d^	2354 ± 105 ^b^	71.0 ± 3.6 ^a^	384 ± 16 ^ab^	665 ± 75 ^a^	27.0 ± 1.7 ^a^
	Common bean	604 ± 39 ^c^	1469 ± 68 ^d^	53.3 ± 5.5 ^cd^	303 ± 1 ^d^	350 ± 3 ^c^	8.0 ± 1.0 ^c^
Star-shaped	Red lentil	n.d.	n.d.	n.d.	n.d.	n.d.	n.d.
	Faba bean	747 ± 21 ^b^	2582 ± 87 ^ab^	59.7 ± 2.5 ^bc^	355 ± 6 ^c^	513 ± 22 ^b^	15.0 ± 2.0 ^b^
	Brown pea	688 ± 13 ^b^	2629 ± 80 ^a^	65.7 ± 2.3 ^ab^	366 ± 9 ^bc^	497 ± 4 ^b^	19.0 ± 1.0 ^b^
	Common bean	1030 ± 41 ^a^	2372 ± 129 ^ab^	50.0 ± 1.0 ^d^	393 ± 8 ^a^	429 ± 20 ^bc^	9.3 ± 1.2 ^c^
Legume		*p* < 0.001	*p* < 0.001	*p* < 0.001	*p* < 0.001	*p* < 0.001	*p* < 0.001
Die		*p* < 0.001	*p* < 0.001	*p* = 0.942	*p* < 0.05	*p* < 0.05	*p* < 0.05
Legume × die		*p* < 0.001	*p* < 0.001	*p* < 0.01	*p* < 0.001	*p* < 0.001	*p* = 0.001

n.d. = not detectable. Different letters in the columns indicate the significant differences (*p* < 0.05) between both shapes of products (sphere and star) considering the interaction between the two factors (*legume* and *die*).

**Table 6 foods-10-03015-t006:** Color parameters (mean ± standard deviation; *n* = 5) of spherical and star-shaped extruded products obtained from different legume flours.

Die Configuration	Legume	Color Parameters		
		*L**	*a**	*b**
Spherical	Red lentil	74.64 ± 0.07 ^b^	4.18 ± 0.03 ^c^	31.88 ± 0.02 ^b^
	Faba bean	70.45 ± 0.21 ^f^	2.75 ± 0.06 ^e^	20.94 ± 0.07 ^f^
	Brown pea	70.46 ± 0.09 ^f^	2.55 ± 0.05 ^f^	19.54 ± 0.03 ^h^
	Common bean	72.64 ± 0.04 ^d^	4.91 ± 0.01 ^a^	24.54 ± 0.05 ^c^
Star-shaped	Red lentil	74.87 ± 0.06 ^a^	4.52 ± 0.03 ^b^	33.24 ± 0.04 ^a^
	Faba bean	72.24 ± 0.06 ^e^	2.22 ± 0.02 ^h^	20.70 ± 0.08 ^g^
	Brown pea	73.25 ± 0.09 ^c^	2.39 ± 0.02 ^g^	21.09 ± 0.07 ^e^
	Common bean	75.04 ± 0.02 ^a^	4.09 ± 0.01 ^d^	24.12 ± 0.02 ^d^
Legume		*p* < 0.001	*p* < 0.001	*p* < 0.001
Die		*p* < 0.001	*p* < 0.001	*p* < 0.001
Legume × die		*p* < 0.001	*p* < 0.001	*p* < 0.001

Different letters in the columns indicate significant differences (*p* < 0.05) between both shapes of products (sphere and star) considering the interaction between the two factors (*legume* and *die*).

**Table 7 foods-10-03015-t007:** Anti-nutritional compounds (mean ± standard deviation; *n* = 3) of spherical and star-shaped extruded products obtained from different legume flours.

Die Configuration	Legume	Phytates Content (mg Phytic Acid/g d.m.)	Oligosaccharide Content (mg/g d.m.)		
			Verbascose	Stachyose	Raffinose
Spherical	Red lentil	3.55 ± 0.05 ^d^	17.02 ± 0.27 ^b^	31.09 ± 0.15 ^d^	14.18 ± 0.14 ^b^
	Faba bean	6.53 ± 0.21 ^b^	13.69 ± 0.06 ^c^	41.45 ± 0.64 ^a^	3.84 ± 0.02 ^f^
	Brown pea	4.47 ± 0.04 ^c^	28.48 ± 0.48 ^a^	22.04 ± 0.38 ^e^	11.43 ± 0.30 ^c^
	Common bean	6.89 ± 0.09 ^a^	12.51 ± 0.42 ^d^	34.76 ± 0.42 ^c^	5.54 ± 0.40 ^e^
Star-shaped	Red lentil	3.55 ± 0.11 ^d^	14.50 ± 0.44 ^c^	30.58 ± 0.67 ^d^	17.43 ± 0.10 ^a^
	Faba bean	6.66 ± 0.22 ^ab^	14.08 ± 0.62 ^c^	42.52 ± 0.07 ^a^	3.67 ± 0.32 ^f^
	Brown pea	4.17 ± 0.03 ^c^	28.70 ± 0.12 ^a^	22.45 ± 0.38 ^e^	11.08 ± 0.04 ^c^
	Common bean	6.98 ± 0.03 ^a^	11.41 ± 0.41 ^d^	37.31 ± 1.04 ^b^	6.93 ± 0.36 ^d^
Legume		*p* < 0.001	*p* < 0.001	*p* < 0.001	*p* < 0.001
Die		*p* = 0.655	*p* < 0.001	*p* = 0.001	*p* < 0.001
Legume × die		*p* < 0.05	*p* < 0.001	*p* < 0.01	*p* < 0.001

Different letters in the columns indicate significant differences (*p* < 0.05) between both shapes of products (sphere and star) considering the interaction between the two factors (*legume* and *die*).

**Table 8 foods-10-03015-t008:** Characterization of extruded products (spheres and stars) by sensory attribute liking.

Die Configuration	Legume	Appearance	Texture	Taste	Aftertaste
Spherical	Red lentil	127 ^bc^*	178 ^b^	167 ^a^	149 ^ab^
	Faba bean	106 ^cd^	98 ^cd^	124 ^b^	121 ^bc^
	Brown pea	107 ^cd^	105 ^cd^	101 ^bc^	113 ^bc^
	Common bean	76 ^d^	75 ^d^	75 ^c^	103 ^c^
Star-shaped	Red lentil	199 ^a^	215 ^a^	199 ^a^	183 ^a^
	Faba bean	160 ^b^	119 ^c^	126 ^b^	121 ^bc^
	Brown pea	132 ^bc^	123 ^c^	116 ^b^	115 ^bc^
	Common bean	101 ^cd^	95 ^cd^	100 ^bc^	102 ^c^

* Sums of rank by 28 semi-trained panelists (the smaller the sum, the better the sensory attribute liking). Different letters in the columns indicate significant differences among the sums of rank for each product (different limit at 5% of 35.93, *z* = 1.96), at *p* < 0.05.

## Data Availability

The data presented in this study are available upon request from the corresponding author.

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
