# Peer review of "Effect of Die Configuration on the Physico-Chemical Properties, Anti-Nutritional Compounds, and Sensory Features of Legume-Based Extruded Snacks"

_foods, 2021, doi:10.3390/foods10123015_

Round 1
Reviewer 1 Report
The manuscript presents interesting information regarding the effect of the extruder die shape on different physicochemical and sensory properties, and anti-nutritional compounds of extruded legume-based snacks.
However, some details must be reviewed before publication.
Page 2; Table 1.
Please add the standard deviation values for fats, carbohydrates, fibers, and proteins.
Please report the methodology used to determine the proximal composition of legumes.
In Table 1, fibers is referred to crude fiber, or dietary fiber?, please describe.
Page 3; line 85
The authors mention:
The extruder, having a 5 kW motor, was equipped with two 38CrMoAl screws (32 mm diameter, 660 mm length, 500 rpm maximum screw speed).
Also, in page 3; lines 85-88
The authors mention:
The operating conditions were as follows: feed rate 2.5 g/s; barrel temperature of the three heating zones = 55, 95, and 125 °C, respectively; die temperature160 °C; feed moisture16 g/100 g; screw speed 23 Hz; cutting speed 19 Hz.
Please standardize the units for screw speed, initially the units are reported in rpm, and later in Hz. In the abstract section of the manuscript, the screw speed is also reported in Hz units.
Page 4; lines 127-128
The authors mention:
………… the suspension was mixed for 15 min, and then centrifuged for 10 min at 6000 rpm to remove the insoluble sediment.
Please report the units in x g (centrifugal forces) instead of rpm, x g units depend on the radius of the rotor.
Page 5; line 201
Please homogenize the way of reporting the significance levels by writing p instead of p in some sections of manuscript.
Page 8; lines 289-291
The authors mention:
…… .., while WSI was higher in the spherical than in circular extrudates, due to the increase of extrusion temperature induced in case of the circular die
Please check the wording of the sentence, and write "star-shaped" instead of "circular".
The sentence could be as follows:
…… ..while WSI was higher in the spherical than in star-shaped extrudates, due to the increase of extrusion temperature induced in case of the spherical die.
Page 10; Table 6
Please verify the color parameter a * for spherical extruded products.
Is it correct that the parameter a * of the common bean (4.91 ± 0.01) is greater than the red lentil (4.18 ± 0.01)?, because in star-shaped product the value of a * of the red lentil (4.52 ± 0.03) is higher than common bean (4.09 ± 0.01), due to its reddish coloration.

Author Response
Response to Reviewer 1
The manuscript presents interesting information regarding the effect of the extruder die shape on different physicochemical and sensory properties, and anti-nutritional compounds of extruded legume-based snacks.
However, some details must be reviewed before publication.
Page 2; Table 1. Please add the standard deviation values for fats, carbohydrates, fibers, and proteins. Please report the methodology used to determine the proximal composition of legumes. In Table 1, fibers is referred to crude fiber, or dietary fiber?, please describe.
Response: Considering that our extruded breakfast snacks were made with 100% legume flour, we only considered the nutritional (macronutrients) characteristics of these flours as listed on their labels (this detail was already reported at line 73). However, in the food labels SD are not reported so we cannot add it, sorry. Usually, labels list the total dietary fibers content. Therefore, we fixed the fiber designation within the table.
Page 3; line 85 The authors mention: The extruder, having a 5 kW motor, was equipped with two 38CrMoAl screws (32 mm diameter, 660 mm length, 500 rpm maximum screw speed). Also, in page 3; lines 85-88 The authors mention: The operating conditions were as follows: feed rate 2.5 g/s; barrel temperature of the three heating zones = 55, 95, and 125 °C, respectively; die temperature160 °C; feed moisture16 g/100 g; screw speed 23 Hz; cutting speed 19 Hz.
Please standardize the units for screw speed, initially the units are reported in rpm, and later in Hz. In the abstract section of the manuscript, the screw speed is also reported in Hz units.
Response: Thanks for noting. Considering that “rpm” is the unit more diffused in the literature, we have converted the units for screw speed from Hz to rpm. You can find these modifications in the abstract (line 20) and in the main text (line 92)
Page 4; lines 127-128 The authors mention: ………… the suspension was mixed for 15 min, and then centrifuged for 10 min at 6000 rpm to remove the insoluble sediment. Please report the units in x g (centrifugal forces) instead of rpm, x g units depend on the radius of the rotor.
Response: We have changed the rpm unit to xg (line 133).
Page 5; line 201 Please homogenize the way of reporting the significance levels by writing p instead of p in some sections of manuscript.
Response: We checked the entire paper and homogenized all the “p”. Lines 213 and 474.
Page 8; lines 289-291 The authors mention: …… .., while WSI was higher in the spherical than in circular extrudates, due to the increase of extrusion temperature induced in case of the circular die. Please check the wording of the sentence, and write "star-shaped" instead of "circular". The sentence could be as follows: …… ..while WSI was higher in the spherical than in star-shaped extrudates, due to the increase of extrusion temperature induced in case of the spherical die.
Response: Thank you for your exhaustive comment. We have changed the word "circular" to "star-shaped". However, considering the geometry of the die, we left "circular" as its adjective at the end of the same sentence. Line 295.
Page 10; Table 6 Please verify the color parameter a * for spherical extruded products. Is it correct that the parameter a * of the common bean (4.91 ± 0.01) is greater than the red lentil (4.18 ± 0.01)?, because in star-shaped product the value of a * of the red lentil (4.52 ± 0.03) is higher than common bean (4.09 ± 0.01), due to its reddish coloration.
Response: We verified the color parameter a*, which is correct. During extrusion-cooking, the red lentil lost its color due to pigment degradation [50], resulting in yellow products. Therefore, the a* parameter of these products could be lower than that of common bean. Furthermore, when comparing red lentil extrudates with other products, we found an opposite trend for the degree of gelatinization. It would appear that the use of the star-shaped die induced overheating of the product obtained from red lentil leading to an increased reddish coloration due to brown compounds produced by Maillard reaction [12].

Reviewer 2 Report
The effect of the die form could be improved by changing extrusion conditions is a basic work when you run a extruder. The effect on texture is easy explained by the effect of pressure difference in the die. Change your objective and therefore dicussion and conclusion
Author Response
Response to Reviewer 2
The effect of the die form could be improved by changing extrusion conditions is a basic work when you run a extruder. The effect on texture is easy explained by the effect of pressure difference in the die. Change your objective and therefore dicussion and conclusion.
Response. Yes, the effect of the die form could be improved by changing extrusion conditions but we kept them constant to quantify the influence of the die alone. Yes, the effect on texture is easy explained by the effect of pressure difference in the die, in fact we commented the results as an effect of different friction which in turn induced a different thermal effect by changing the die configuration (shape and cross section area). The effect of different extrusion conditions has been already evaluated in a previous paper by the same research group (Pasqualone, A., Costantini, M., Labarbuta, R. and Summo, C., 2021. Production of extruded-cooked lentil flours at industrial level: Effect of processing conditions on starch gelatinization, dough rheological properties and techno-functional parameters. LWT, 147, p.111580) as well as by other authors. The effect of die, instead, in terms of shape and cross-section area, had not been considered by the studies reported in the scientific literature so far, despite being of great practical interest.

Reviewer 3 Report
The manuscript „ Effect of die on the physico-chemical properties, anti-nutritional compounds, and sensory features of legume-based extruded snacks“ is well designed and written, and has important information, followed by explanations for the impact of the effect of two different dies on the physico-chemical properties, anti-nutritional compounds, and sensory features of extruded breakfast snacks prepared from 100% legume flour.
There are just few issues to be addressed:
Line 35- 36
Sentence (line 35-36) should be revised. It is the same as in abstract.
‘’However, legumes are not valued by all consumers, mostly due to the prolonged soaking and cooking process they require’’.
According to line 45-48
In section Extrusion-cooking process the authors should explain why in their study use just specified extrusion parameters?
How do you decide to follow these extrusion cooking conditions: 2.5 g/s feed rate; 160±1 °C die temperature; 16±1 g/100 g feed moisture; 23 19 Hz screw speed; 19 Hz cutting speed? Do you have some preliminary optimization process conditions?
Line 87-88
Please convert screw speed units from Hz to rpm
Line 148-152
Please add more details in differences between measurement crispness and crunchiness?
Line 277-278
Is negative correlation between ER and BD was observed in your study? Please specify?
Line 282-291
Explanations are to general; please specify how extrusion cooking temperature in your case influences WAI and WSI. I don’t see that you follow influence of extrusion cooking temperature on these two output parameters. Furthermore, taking into account constant temperature in your barrel there are no need for explanation of influence temperature on these two outputs.
Line 300-309
Results should be discussed more extensively in term of effect of type die on crispness and crunchiness. What is the main difference between these two outputs and reason for determine bout of them?
Line 308- 309
Crispness strongly influences the consumer acceptability of extruded products [42]. Why is important to determine crunchiness, is it important for consumer acceptability of extruded products?
Line 310-311
In Table 5 is there any unit for crispness? How do you get those values?
Line 314- 316
Do you have results of specific mechanical energy production influence on hardness and crispness? Do you have a data for mechanical energy?

Author Response
Response to Reviewer 3
The manuscript „ Effect of die on the physico-chemical properties, anti-nutritional compounds, and sensory features of legume-based extruded snacks“ is well designed and written, and has important information, followed by explanations for the impact of the effect of two different dies on the physico-chemical properties, anti-nutritional compounds, and sensory features of extruded breakfast snacks prepared from 100% legume flour.
There are just few issues to be addressed:
Line 35- 36 Sentence (line 35-36) should be revised. It is the same as in abstract.
‘’However, legumes are not valued by all consumers, mostly due to the prolonged soaking and cooking process they require’’.
Response: The sentence has been revised (lines 35-37).
According to line 45-48 In section Extrusion-cooking process the authors should explain why in their study use just specified extrusion parameters?
Response: The purpose of this study was to evaluate the effect of two different dies on the the physico-chemical properties, anti-nutritional compounds, and sensory characteristics of expanded legume snacks, so their production was carried out after an optimization, through preliminary tests, of process conditions. We have specified it at line 90.
How do you decide to follow these extrusion cooking conditions: 2.5 g/s feed rate; 160±1 °C die temperature; 16±1 g/100 g feed moisture; 23 19 Hz screw speed; 19 Hz cutting speed? Do you have some preliminary optimization process conditions?
Response: As above specified, the experimental trials were carried out after an optimization, through preliminary tests, of process conditions. We have specified it at line 90.
Line 87-88 Please convert screw speed units from Hz to rpm
Response: Thank you for your comment. We have converted the units for screw speed from Hz to rpm in the abstract (line 20) and in the main text (line 92).
Line 148-152 Please add more details in differences between measurement crispness and crunchiness?
Response: We have added more details about crispness and crunchiness, and you can find them at lines 157-162.
Line 277-278 Is negative correlation between ER and BD was observed in your study? Please specify?
Response: We have specified (lines 286-287) the negative correlation found between ER and BD adding the Pearson correlation coefficient (r) and the p-value.
Line 282-291 Explanations are to general; please specify how extrusion cooking temperature in your case influences WAI and WSI. I don’t see that you follow influence of extrusion cooking temperature on these two output parameters. Furthermore, taking into account constant temperature in your barrel there are no need for explanation of influence temperature on these two outputs.
Response: Despite we have worked with constant extrusion-cooking temperatures, the two types of the die (circular and star-shaped), with different cross-sections (19.6 vs. 35.9 mm2), indiced a different thermal effect. Considering that the circular cross-section was smaller than the star-shaped one, conditioned flour was subjected to more elevated friction and pressure as flowing through the circular die, than through the star-shaped one. In turn, higher friction induced heat generation and increased the actual extrusion temperature. Therefore, as other parameters, WAI and WSI could have been affected by this temperature difference [41]. You can find this explanation at lines 306-317. Furthermore, we have semplified and partly deleted the sententes at lines 317-325.
Line 300-309 Results should be discussed more extensively in term of effect of type die on crispness and crunchiness. What is the main difference between these two outputs and reason for determine bout of them?
Response: Thanks for noting. A more detailed explanation on the difference between these two outputs has been added at lines 343-358. Crispness and crunchiness are two different sensations that in the human brain are induced by different stimuli during the dynamic process of mastication [46]. Crispness could be identified as the perceived force necessary to separate the product into two or more distinct pieces during a single bite with the incisors [46]. Crisp products are characterised by a brittle and low-density structure, which easily breaks and generates loud and high-pitched sounds when fractured [47].
Line 308- 309 Crispness strongly influences the consumer acceptability of extruded products [42]. Why is important to determine crunchiness, is it important for consumer acceptability of extruded products?
Response: Yes, consumer acceptability is strongly influenced also by crunchiness [43]. Crunchiness is an attribute that identifies harder foods that emit sounds at lower frequencies than crisp foods; while crispiness is used for products characterised by a brittle and low density structure, which easily breaks and generates loud and high-pitched sounds when fractured [43]. Therefore, it is important to determine crunchiness separately from crispness. Crispness and crunchiness are both important quality attributes used to describe the texture of extruded snack products.
Line 310-311 In Table 5 is there any unit for crispness? How do you get those values?
Response: There is not any unit for crispness because it was measured by counting the positive peaks in the texture analyzer graph. In fact, this parameter is expressed as the number of positive peaks [24].
Line 314- 316 Do you have results of specific mechanical energy production influence on hardness and crispness? Do you have a data for mechanical energy?
Response: Thank you for your comment, but unfortunately we did not record the motor torque. Therefore, it is not possible to calculate the SME.
Conclusions are written briefy.
Response: The conclusions were extended by including more information about the results (lines 489-500).
Considering all, I suggest this paper to be published in Foods after minor revision.

Reviewer 4 Report
The work is very well presented, the results in the same way, well presented and the discussion is very complete. I annex the observations of the review.
General observations:
Materials and methods
Table 1 are results, transfer it to the corresponding section.
Results and Discussion
Page 8, lines 276-280. The authors mention that there is an inversely proportional relationship between ER and BD that justifies that the red lentil presents lower extrusion characteristics, the presence of fiber will have an effect on WAI, WSI, and DG ?, if the fiber has an effector on extrusion, why with brown pea flour is there greater expansion than with bean flour?
Page 9, lines 296-309. What factors determine or have a greater impact on the crispiness of the brown pea? Why is this characteristic not maintained in bowl life?
Page 11, lines 344-356. The observed darkening of the final product is probably due more to the presence of pigments than to the Maillard reactions since the protein-carbohydrate ratio mentioned in table 1 does not correspond to the darkening results obtained.
Lines 357-371. To improve the discussion, I suggest mentioning the reduction percentages in each case of the antinutritional compounds.
Author Response
Response to Reviewer 4
The work is very well presented, the results in the same way, well presented and the discussion is very complete. I annex the observations of the review.
General observations:
Materials and methods
Table 1 are results, transfer it to the corresponding section.
Response: The nutritional characteristics of the flours used were taken from their labels (this detail was already reported at line 73), therefore they are not results from the present investigation. For this reason, it is more appropriate to keep them in the Materials and methods section, as an additional information which could be given in parenthesis when listing the materials used, but was given as a table for allowing easier reading.
Results and Discussion
Page 8, lines 276-280. The authors mention that there is an inversely proportional relationship between ER and BD that justifies that the red lentil presents lower extrusion characteristics, the presence of fiber will have also an effect on WAI, WSI, and DG ?, if the fiber has an effect on extrusion, why with brown pea flour is there a greater expansion than with bean flour?
Response. We have expanded the discussion on the effect of fibres at lines 308-310; 312-317. Regarding brown pea flour, we checked the values in the tables and it had a lower fiber content than common bean flour, therefore it had higher ER than the latter.
Page 9, lines 296-309. What factors determine or have a greater impact on the crispiness of the brown pea? Why is this characteristic not maintained in bowl life?
Response. The highest expansion ratio and the lowest bulk density and hardness values, which characterized brown pea products, are the factors with a greater impact on the crispiness of the brown pea products. We added a specific sentence at lines 340-341. Furthermore, this trend was also maintained by brown pea products in bowl life, as reported in table 5.
Page 11, lines 344-356. The observed darkening of the final product is probably due more to the presence of pigments than to the Maillard reactions since the protein-carbohydrate ratio mentioned in table 1 does not correspond to the darkening results obtained.
Response. We added a sentence to take into account the influence of the pigments at line 386-387.
Lines 357-371. To improve the discussion, I suggest mentioning the reduction percentages in each case of the antinutritional compounds.
Response: We mentioned the variation percentages of antinutritional compounds during extrusion-cooking, as suggested. You will find them at lines 434-437; 444-449.

Reviewer 5 Report
The study by Costantini et al. assessed select physical and chemical characteristics of extrudates made from lentil flours. In my opinion, this topic is of interest to readers from academia and industry, and the study design is appropriate. The paper is well-written and diligently formatted. Overall, it is a solid paper. I therefore recommend publication after minor revisions.
My main question to the authors is this: For the results shown in Tables 4 - 7, a two-way ANOVA was performed to assess the influence of die configuration, lentil type, and their interactions. As almost all parameters were influenced by the factors, why not block the analysis according to factors? It seems to me that this would have been the better strategy.
Aside from this, I mostly have minor suggestions, most of which are about the writing style:
I think it would be better to include a term like "configuration" or "type" after "die" in the title (e.g., Effect of die type on the ...). I would also add "type" or "configuration", or an equivalent term, to the table legends, e.g., for table 4.
L31: I suggest replacing "to" with "for"
L36: I would replace "elaborating" with "developing"
L42: Since "snacks" is plural, also put shape, texture and colour in plural form
L49: Replace "researches" with "studies"
Table 1: Change "Fibers" to "Dietary fiber" - I assume that this was what the manufacturers of these legume flours reported.
L170: It would sound better to remove "the" from this header
L254: Remove "resulted"
L278-281: Please briefly discuss why exactly dietary fiber restricts expansion, and insert some relevant references.
L333: Either remove "the" before "extrusion-cooking" or add keep the article and add "process" afterwards
L401: I would add "being" before "difficult to chew"; question regarding "with neutral taste and aftertaste" part of the sentence: If the taste properties of red lentil extrudates were neutral in nature, did they really contribute to the lowest ranking? It would seem to me that a neutral taste would not improve their liking, but also not decrease it...or did the panelists think that they were too bland and boring? Maybe this could be made clearer
L409: I would suggest replacing "good" with another term, because it is highly subjective. Same for L436
L421-423: Would it work to add "used in industry" after "dies", and "on legumes" after "studied"?
L429: I suggest adding "by panelists" or something equivalent after "appreciated"
L433: The authors did not evaluate the effect of dietary fiber in detail. It is a reasonable assumption, but since no work was carried out related to it in this study, I suggest replacing "mostly" with "presumably" or an equivalent term.
Author Response
Response to Reviewer 5
The study by Costantini et al. assessed select physical and chemical characteristics of extrudates made from lentil flours. In my opinion, this topic is of interest to readers from academia and industry, and the study design is appropriate. The paper is well-written and diligently formatted. Overall, it is a solid paper. I therefore recommend publication after minor revisions.
My main question to the authors is this: For the results shown in Tables 4 - 7, a two-way ANOVA was performed to assess the influence of die configuration, lentil type, and their interactions. As almost all parameters were influenced by the factors, why not block the analysis according to factors? It seems to me that this would have been the better strategy.
Response: Thank you for your suggestion, but not all parameters were affected by the factors considered in the two-way ANOVA; in fact, WAI (Table 4), crispness (Table 5), and total phytates (Table 7) were not affected by the different conformation of the die. Therefore, we considered it was appropriate to perform the two-way ANOVA considering both factors (die and legume) and their interaction (legume*die) to understand how parameters vary when these factors vary.
Aside from this, I mostly have minor suggestions, most of which are about the writing style:
I think it would be better to include a term like "configuration" or "type" after "die" in the title (e.g., Effect of die type on the ...). I would also add "type" or "configuration", or an equivalent term, to the table legends, e.g., for table 4.
Response: Thanks for your input. Actually, we already used the words “die configuration” in the Conclusions. We added “configuration” to the title, aims of the work and Tables 4-8.
L31: I suggest replacing "to" with "for"
Response: We amended the sentence as suggested.
L36: I would replace "elaborating" with "developing"
Response: We amended the sentence as suggested.
L42: Since "snacks" is plural, also put shape, texture and colour in plural form
Response: We amended the sentence as suggested.
L49: Replace "researches" with "studies"
Response: We amended the sentence as suggested.
Table 1: Change "Fibers" to "Dietary fiber" - I assume that this was what the manufacturers of these legume flours reported.
Response: We amended the table as suggested.
L170: It would sound better to remove "the" from this header
Response: We amended the sentence as suggested.
L254: Remove "resulted"
Response: We amended the sentence as suggested.
L278-281: Please briefly discuss why exactly dietary fiber restricts expansion, and insert some relevant references.
Response: Dietary fibres lead to cell wall rupture before air bubbles could expand reducing the overall expansion (Liu, Y., Hsieh, F., Heymann, H., & Huff, H. E. (2000). Effect of process conditions on the physical and sensory properties of extruded oat‐corn puff. Journal of food science, 65(7), 1253-1259). As a result, extruded products with a high fibre content are usu-ally compact, hard, not crispy, and have undesirable texture (Lue, S., Hsieh, F., & Huff, H. E. (1991). Extrusion cooking of corn meal and sugar beet fiber: effects on expansion properties, starch gelatinization, and dietary fiber content. Cereal Chemistry, 68(3), 227-234). This trend was reported by many researchers (Liu, Y., Hsieh, F., Heymann, H., & Huff, H. E. (2000). Effect of process conditions on the physical and sensory properties of extruded oat‐corn puff. Journal of food science, 65(7), 1253-1259; Shirazi, S. L., Koocheki, A., Milani, E., & Mohebbi, M. (2020). Production of high fiber ready-to-eat expanded snack from barley flour and carrot pomace using extrusion cooking technology. Journal of Food Science and Technology, 1-13; Pérez‐Navarrete, C., Gonzalez, R., Chel‐Guerrero, L., & Betancur‐Ancona, D. (2006). Effect of extrusion on nutritional quality of maize and Lima bean flour blends. Journal of the Science of Food and Agriculture, 86(14), 2477-2484). We expanded the explanation about the effect of fibre content at lines 296-299.
L333: Either remove "the" before "extrusion-cooking" or keep the article and add "process" afterwards
Response: Thanks, we deleted “the” before “extrusion-cooking”.
L401: I would add "being" before "difficult to chew"; question regarding "with neutral taste and aftertaste" part of the sentence: If the taste properties of red lentil extrudates were neutral in nature, did they really contribute to the lowest ranking? It would seem to me that a neutral taste would not improve their liking, but also not decrease it...or did the panelists think that they were too bland and boring? Maybe this could be made clearer
Response: Thanks, we amended the sentence as suggested.
L409: I would suggest replacing "good" with another term, because it is highly subjective. Same for L436
Response: Thanks for suggestion. We amended the sentence.
L421-423: Would it work to add "used in industry" after "dies", and "on legumes" after "studied"?
Response: We amended the sentence as suggested.
L429: I suggest adding "by panelists" or something equivalent after "appreciated"
Response: We amended the sentence as suggested.
L433: The authors did not evaluate the effect of dietary fiber in detail. It is a reasonable assumption, but since no work was carried out related to it in this study, I suggest replacing "mostly" with "presumably" or an equivalent term.
Response: We amended the sentence as suggested.
